# Fine-Tuning of Sequence Specificity by Near Attack Conformations in Enzyme-Catalyzed Peptide Hydrolysis

**S. Kashif Sadiq** [1,2]

1    Heidelberg Institute for Theoretical Studies, Schloss-Wolfsbrunnenweg 35, 69118 Heidelberg, Germany; kashif.sadiq@h-its.org or syedkashifsadiq@gmail.com
2    Infection Biology Unit, Universitat Pompeu Fabra, Carrer Doctor Aiguader 88, Barcelona Biomedical Research Park, 08003 Barcelona, Spain

**Abstract:** The catalytic role of near attack conformations (NACs), molecular states that lie on the pathway between the ground state (GS) and transition state (TS) of a chemical reaction, is not understood completely. Using a computational approach that combines Bürgi–Dunitz theory with all-atom molecular dynamics simulations, the role of NACs in catalyzing the first stages of HIV-1 protease peptide hydrolysis was previously investigated using a substrate that represents the recognized SP1-NC cleavage site of the HIV-1 Gag polyprotein. NACs were found to confer no catalytic effect over the uncatalyzed reaction there ($\Delta\Delta G_N^{\ddagger} \sim 0$ kcal/mol). Here, using the same approach, the role of NACs across multiple substrates that each represent a further recognized cleavage site is investigated. Overall rate enhancement varies by $|\Delta\Delta G^{\ddagger}| \sim 12$–$15$ kcal/mol across this set, and although NACs contribute a small and approximately constant barrier to the uncatalyzed reaction ($<\Delta G_N^{\ddagger u}> = 4.3 \pm 0.3$ kcal/mol), they are found to contribute little significant catalytic effect ($|\Delta\Delta G_N^{\ddagger}| \sim 0$–$2$ kcal/mol). Furthermore, no correlation is exhibited between NAC contributions and the overall energy barrier ($R^2 = 0.01$). However, these small differences in catalyzed NAC contributions enable rates to match those required for the kinetic order of processing. Therefore, NACs may offer an alternative and subtle mode compared to non-NAC contributions for fine-tuning reaction rates during complex evolutionary sequence selection processes—in this case across cleavable polyproteins whose constituents exhibit multiple functions during the virus life-cycle.

**Keywords:** enzyme catalysis; bimolecular reactions; molecular dynamics; HIV-1 protease; near attack conformations; enzyme specificity

## 1. Introduction

According to Pauling [1], rate enhancement due to enzyme catalysis stems from the tighter binding of the transition state (TS) as compared to the ground state (GS) of a substrate undergoing chemical reaction. This is thermodynamically equivalent to a negative activation free energy difference ($\Delta\Delta G^{\ddagger}$) between the catalyzed ($\Delta G^{\ddagger c}$) and the uncatalyzed ($\Delta G^{\ddagger u}$) reaction where:

$$\Delta\Delta G^{\ddagger} = \Delta G^{\ddagger c} - \Delta G^{\ddagger u}. \tag{1}$$

Activation rates are describable within a generalized transition state theory (TST) framework [2] which extends the original formulation of TST [3] such that the rate of a chemical reaction is defined as:

$$k(T) = \gamma(T)(k_B T/h)(C^0)^{1-n} exp[-\Delta G^{\ddagger}/RT], \tag{2}$$

where $T$ is the temperature, $R$ is the gas constant, $C^0$, the standard state concentration of the reactant, $n$ the order of the reaction, $\Delta G^{\ddagger}$ is the standard-state free energy of activation, and $\gamma(T)$ is the generalized transmission coefficient. Rate enhancement via enzyme catalysis can then be afforded by lowering $\Delta G^{\ddagger}$ and/or raising $\gamma(T)$. The molecular origin of rate increase can stem from a number of processes such as quantum mechanical tunneling [4–6], non-equilibrium dynamics [7–12], changes in substrate flexibility between the TS and the GS [13–15], preorganization of electrostatic interactions that favor formation of the TS [16–22] and the thermodynamic stabilization of near attack conformations (NACs)—GS conformations that lie on the transition path closer to the TS [13,23–30].

The role of NACs in catalyzing enzymatic reactions has recently been examined for nucleophilic bimolecular reactions [31], specifically enzyme-catalyzed peptide hydrolysis by HIV-1 protease. The catalyzed reaction proceeds via a likely general-acid/general base (GA/GB) mechanism [32] that forms a gem-diol intermediate in a two-step process. The first step constitutes the largest energy barrier [33] and involves nucleophilic water attack of the lytic peptide bond carbon atom. This is facilitated by hydrogen bonding between the monoprotonated D25 residue in the catalytic aspartic dyad (D25/D25′) with the adjacent carbonyl oxygen and between D25′ and the nucleophilic water (Figure 1). Thus, this step may make use of NACs along the pathway to form a primary transition state (TS). NAC formation was previously studied by developing an all-atom explicit solvent molecular dynamics simulation framework [31] in conjunction with the Bürgi–Dunitz theory [34–37]. This enables identification of NACs in terms of a nucleophilic attack distance threshold ($d_a \leq 3.2$ Å) and a narrow attack angle range ($100° \leq \alpha \leq 110°$) with respect to the relevant carbonyl group (Figure 1). The activation barrier ($\Delta G^{\ddagger}$) can then be separated into NAC ($\Delta G^{\ddagger}_N$) and non-NAC ($\Delta G^{\ddagger}_{nN}$) components:

$$\Delta G^{\ddagger} = \Delta G^{\ddagger}_N + \Delta G^{\ddagger}_{nN}. \tag{3}$$

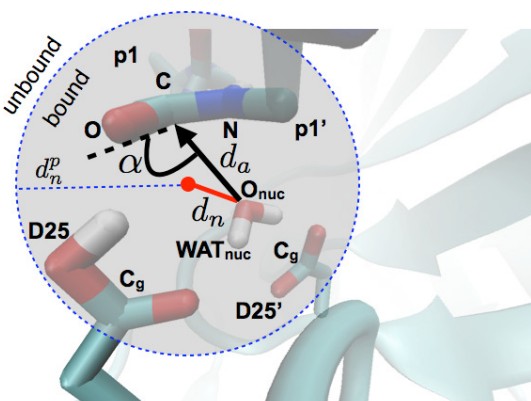

**Figure 1.** Representation of a near attack conformation (NAC) in HIV-1 protease by a nucleophilic water molecule, as described previously [31]. NACs are characterized in terms of Bürgi–Dunitz criteria corresponding to a nucleophile attack angle (WAT$_{nuc}$:O$_{nuc}$-p1:C-p1:O) range ($100° \leq \alpha \leq 110°$) and distance (WAT$_{nuc}$:O$_{nuc}$-p1:C) threshold ($d_a \leq 3.2$ Å). The red dot represents the catalytic center (geometric center of p1:C, p1′:N, D25:CG and D25′:CG), $d_n$ represents the nearest water distance to the catalytic center. The ground state corresponds to a nucleophilic water molecule within the bound perimeter (dashed blue circle) in the catalytic site ($d_n \leq d_n^p$).

The catalytic contribution of any component (X) of the activation barrier is then:

$$\Delta \Delta G^{\ddagger}_X = \Delta G^{\ddagger c}_X - \Delta G^{\ddagger u}_X. \tag{4}$$

Multiple sequence specificity, high selectivity, and finely-tuned rate variation are significant phenomena across biochemical regulation processes and peptide hydrolysis by HIV-1 protease is an excellent example of this. HIV-1 Gag and GagPol precursor polyproteins consist of multiple protein

domains sequentially connected via peptide bonds at their inter-protein junctions. Gag consists of matrix (MA), capsid (CA), spacer peptide 1 (SP1), nucleocapsid (NC), spacer peptide 2 (SP2), and protein p6. Additionally, GagPol is synthesized via a -1 frameshift with a ratio of 1:20 compared to Gag [38], which allows continued sequential translation after NC into a trans-frame-region (TFR) and then precursor Pol that contains the viral protease (PR), reverse-transcriptase (RT and RH), and integrase (IN). The viral protease recognizes and cleaves multiple sequences within Gag and GagPol [39], notably at junctions: MA-CA, CA-SP1, SP1-NC, NC-SP2, SP2-p6, TFR-PR, PR-RT, RT-RH, and RH-IN.

Turnover number, $k_{cat}$, has been experimentally studied widely for HIV-1 protease for various substrates that constitute these sequences—and shows a well-defined rate ordering that ranges over 100-fold from slowest to fastest [39]. Nonetheless, the molecular origin of such differences remains not completely understood.

It has been found that for the enzyme reaction catalyzed by HIV-1 protease bound to the SP1-NC (alternatively, termed p2-NC) octapeptide cleavage substrate and a catalytic water molecule [31] that NACs do not contribute to rate enhancement ($\Delta\Delta G_N^{\ddagger c} \sim 0$ kcal/mol), whilst non-NAC contributions account for ($\Delta\Delta G_{nN}^{\ddagger c} \sim -15$ kcal/mol). As transmission effects are minor (up to $\gamma \sim 10^3$), electrostatic preorganization of the active site is likely the major component of rate enhancement in HIV-1 protease at this cleavage junction.

Nonetheless, the role of NAC thermodynamics in affecting catalytic rate changes amongst differentially recognized substrates has not hitherto been examined. NACs do pose a small but significant barrier in both the catalyzed and uncatalyzed reactions for the SP1-NC cleavage junction [31]. Therefore, it is hypothesized that by exhibiting variation across different recognized substrates NACs may play some role in controlling enzyme specificity through changes in $k_{cat}$. This hypothesis is evaluated here by comparing experimentally determined and estimated $\Delta G^{\ddagger}$ with computed $\Delta G_N^{\ddagger}$ and derived $\Delta G_{nN}^{\ddagger}$ values across the range of nine above-mentioned cleavage junctions with varying $k_{cat}$, through a previously established computational approach that makes use of explicit solvent molecular dynamics simulations to compute mole fractions of NAC-formation during peptide hydrolysis in both the enzyme-bound and enzyme-free substrate systems [31].

## 2. Results

### 2.1. Differential Nucleophilic Water Binding

Computation of the free energy of NAC-formation from a ground state (GS) first requires a suitable definition of the GS for the nucleophilic water molecule. In order to define this, the distance ($d_n$) of the nearest water molecule to the catalytic center (Figure 1) was calculated across the ensemble of MD simulations for each of the enzyme-bound substrate systems.

The density distributions of $d_n$ reveal mostly two distinct bell-shaped regions of non-zero density for each system, corresponding to a state where a nucleophilic water molecule is bound (NWB) in the catalytic site and to when a nucleophilic water molecule is unbound (NWU). The latter density peak is due to the presence of a previously characterized [31] structural non-nucleophilic water molecule that becomes the nearest water molecule to the catalytic center when the nucleophilic water molecule leaves the catalytic site (Figure 2). These two regions are sharply partitioned by a region of near-zero density (dashed black line) for all systems. Thus, a nearest water density description within such a partition threshold ($d_n^p$) is a suitable definition of the GS for a nucleophilic water molecule in each enzyme–substrate complex (Table 1).

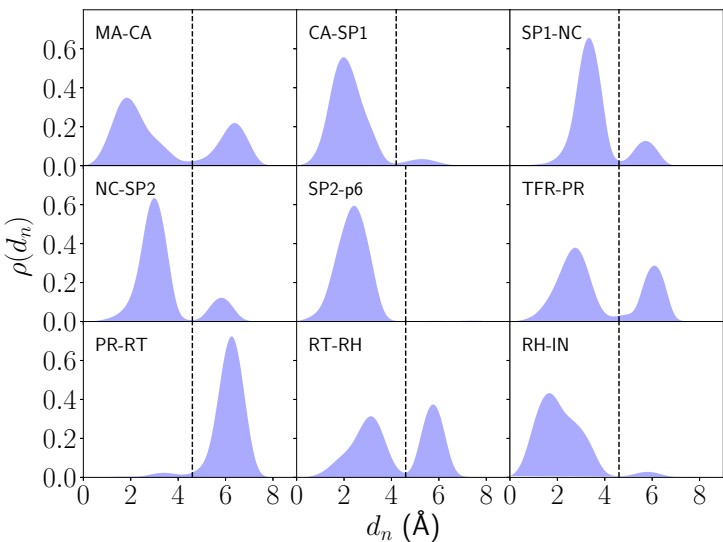

**Figure 2.** Normalized density distributions ($\rho(d_n)$) of the nearest water distance ($d_n$) to the catalytic center across all enzyme-bound substrate systems. The bound-unbound partition distance ($d_n^p$) is shown (dashed black line).

**Table 1.** Free energies of nucleophilic water binding ($\Delta G_{nwb}$) in the enzyme-bound substrate complexes based on assigned partition thresholds, $d_n^p$. All energies are in kcal/mol.

| System | Cleavage Sequence | $d_n^p$ (Å) | $\Delta G_{nwb}$ |
|---|---|---|---|
| MA-CA | SGNY-PIVQ | 4.6 | –0.4 ± 0.1 |
| CA-SP1 | ARVL-AEAM | 4.2 | –1.9 ± 0.5 |
| SP1-NC | ATIM-MQRG | 4.6 | –1.0 ± 0.1 |
| NC-SP2 | RQAN-FLGK | 4.6 | –1.1 ± 0.3 |
| SP2-p6 | PYNF-LQSR | 4.6 | –3.7 ± 1.2 |
| TFR-PR | SFNF-PQIT | 4.6 | –0.4 ± 0.2 |
| PR-RT | TLNF-PISP | 4.6 | 2.0 ± 0.3 |
| RT-RH | AETP-YVDG | 4.6 | –0.1 ± 0.2 |
| RH-IN | RKIL-FLDG | 4.6 | –2.2 ± 0.7 |

However, the profiles of the distributions for different systems do vary. The MA-CA, CA-SP1, SP1-NC, NC-SP2, and RH-IN distributions share a similar profile: the NWB and NWU states are characterized by a single major peak between ~1.5–3.5 Å and a single minor peak between ~5–6.5 Å, respectively. The TFR-PR and RT-RH distributions exhibit a single broad peak at 2.7 Å and 3.1 Å respectively corresponding to the NWB state and one of similar density at 6.1 Å and 5.8 Å respectively, corresponding to NWU. The SP2-p6 system exhibits a single large NWB-peak at 2.4 Å but only trace densities of the NWU state—nonetheless, these constitute two very small peaks at 5.8 Å and 7.4 Å. Thus, for the SP2-p6 system, the nucleophilic water molecule almost never escapes the catalytic site, whilst there are also rare instances when there is neither a water molecule in the GS nor a structural water molecule between the ligand and the protease. Finally, the PR-RT system is the only one for which the NWU state has a significantly higher density than the NWB state. A minor NWB-peak is exhibited at 3.4 Å with an additional trace density peak at 1.8 Å. A much larger peak corresponding to the NWU state is exhibited at 6.3 Å.

Three distinct thermodynamic groups emerge (Table 1) for the free energy of nucleophilic water binding ($\Delta G_{nwb}$). For the MA-CA, SP1-NC, NC-SP2, TFR-PR, and RT-RH systems, $\Delta G_{nwb}$ ranges between –0.1 to –1.1 kcal/mol. Despite the error of the calculation being relatively small in these cases ($\leq$0.3 kcal/mol),

these values are not significantly differentiable from thermal noise, $k_b T \sim 0.6$ kcal/mol at 300 K. A second group corresponds to significantly attractive binding thermodynamics, consisting of CA-SP1 ($-1.9 \pm 0.5$ kcal/mol), SP2-p6 ($-3.7 \pm 1.2$ kcal/mol) and RH-IN ($-2.2 \pm 0.7$ kcal/mol). These systems exhibit favorable binding beyond thermal noise. By contrast, the PR-RT system alone exhibits unfavorable binding association beyond thermal noise ($2.0 \pm 0.3$ kcal/mol).

However, care has to be taken when interpreting these binding free energies quantitatively, especially those with significant favorable or unfavorable values. This is because the frequency of reversible transitions between bound and unbound states varies across the system, implying that an approximate equilibrium ensemble may not be exhibited for every system. Therefore, a simple mole fraction of each state may not be a sufficient descriptor of the free energy of water binding. For example, the favorable $\Delta G_{nwb}$ of CA-SP1, SP2-p6 and RH-IN systems are likely to be overestimated as unbinding of the nucleophilic water molecule is rare even with the 10 µs of aggregate sampling performed here. This is indicative of a significantly higher kinetic barrier for water dissociation, but unlikely to the extent that the thermodynamics would be unfavorable. Similarly, the unfavorable $\Delta G_{nwb}$ of the PR-RT system is also likely to be overestimated; rapid dissociation of the water molecule is exhibited during equilibration, thus production simulations are initiated in a water unbound state and only rare events of subsequent water binding are observed.

Nonetheless, despite these limitations, the partitioning of the nucleophilic water states into unbound and bound remains well-defined, thus the bound sub-ensemble of each system provides an accurate representation of the GS from which subsequent mole fractions of desired states, including NACs, can be computed.

Similarly, the GS of the enzyme-free substrate systems could also be characterized in terms of a distance metric ($d_o$)—in this case, the distance of the nearest water molecule to the midpoint of the lytic peptide bond. The density distributions of $d_o$ expectedly reveal a single-peaked bell-shaped distribution between 3.6–4.2 Å for all systems decaying rapidly and with near-zero density ($\rho(d_o) \leq 0.001$) beyond 6 Å (Figure 3). A threshold of $d_o^p = 6$ Å was therefore set to partition bound from unbound regions—therefore, in the enzyme-free substrate systems, it can be considered that there is always a nucleophilic water molecule proximal to the lytic peptide bond and thus effectively 'bound' in the GS.

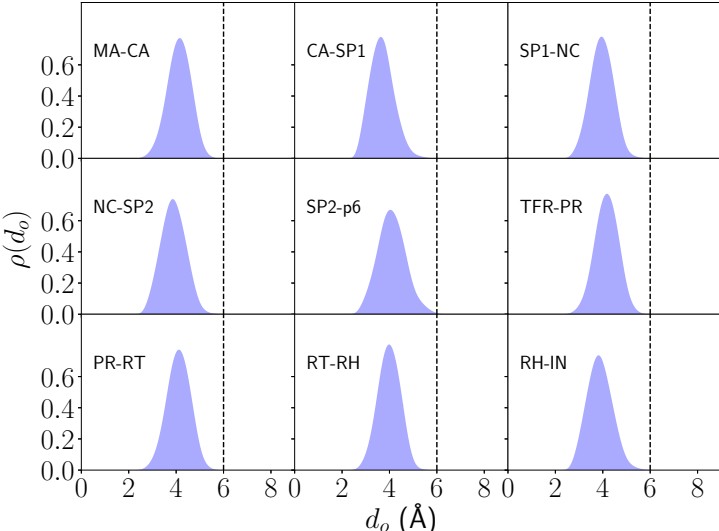

**Figure 3.** Normalized density distributions ($\rho(d_o)$) of the nearest water distance ($d_o$) to the midpoint of the lytic peptide bond across all enzyme-free substrate systems. The bound-unbound partition distance ($d_o^p$), chosen as 6 Å, is shown (dashed black line).

## 2.2. Analysis of the Hydrogen Bond Network

Four distinct hydrogen bonds ($hb_1$-$hb_4$) have previously been characterized in the interaction between the catalytic aspartic acid dyad, the nucleophilic water molecule, and the peptide substrate [31] (Figure 4A). However, this gives rise to a hydrogen bond network of potentially 16 states, when combined, it was previously shown for the SP1-NC system that only 12 of these were ever populated (Figure 4B)—due to the mutual exclusivity of $hb_1$ and $hb_2$. In order to explore the role of hydrogen bonding across the various peptide systems, the population density distribution of each putative state within the hydrogen bond network was computed for the NAC state and the ground state (GS) in general and compared with the network in the unbound state (Figure 4C). Population densities are reported in terms of the potential of mean force (*G* in kcal/mol) derived from the relative mole fractions of each hydrogen bond state, normalized with respect to the density of the given conformation in each system—thus the row vectors in each system sum to unity in terms of population density.

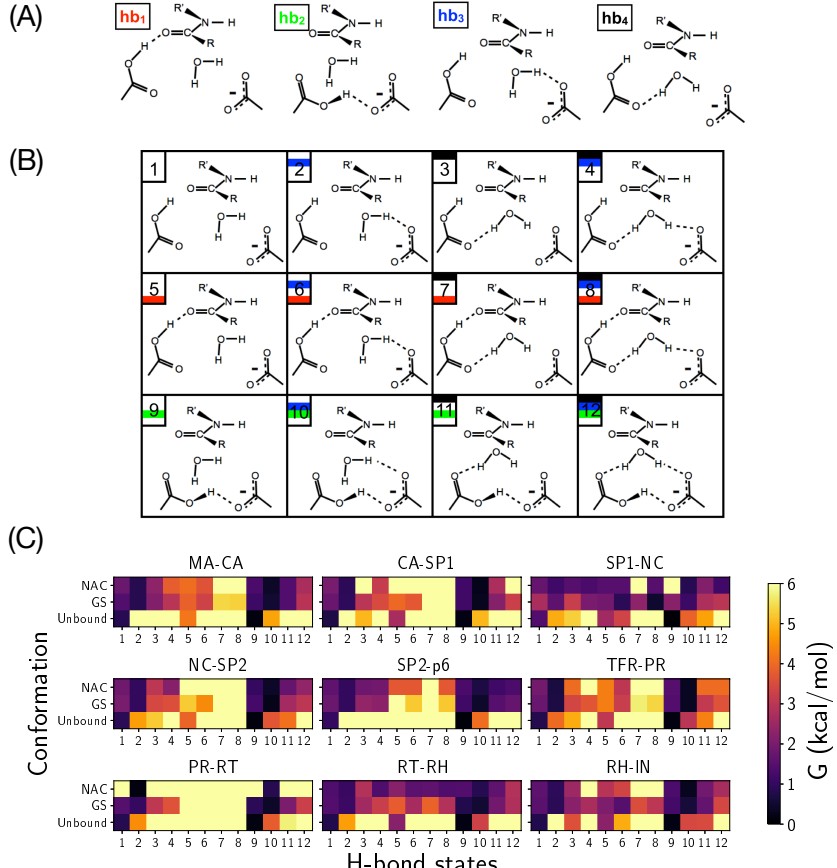

**Figure 4.** (**A**) Four distinct hydrogen bonds that occur in the HIV-1 protease catalytic site when a nucleophilic water molecule is bound and (**B**) twelve possible combinations of hydrogen bond states that exhibit non-zero density as represented in a previous study [31]. (**C**) Thermodynamic variation of catalytic site hydrogen bond state distribution across different substrates when the nucleophilic water molecule is in the NAC, GS, and unbound states.

Only the same 12 hydrogen bond states as reported before [31] are ever populated across all nine systems—therefore, mutual exclusivity of $hb_1$ and $hb_2$ is preserved across all systems. For all systems, the most densely populated hydrogen bond states when the nucleophilic water molecule is unbound are states 1 (no hydrogen bond) and 9 (only the inter-aspartyl hydrogen bond—$hb_2$). By comparison, hydrogen bond state 5, involving $hb_1$, between protonated D25 and the peptide bond carbonyl, is less densely populated in the absence of a bound water molecule. As expected, states involving hydrogen

bonding with a bound water molecule are not exhibited except rare events involving hb$_2$- or hb$_3$-type bonding when the water molecule is at the periphery of the binding site.

The principal change in the hydrogen bond distribution when going from the unbound to the GS is a substantial shift in density towards state 2 (between water and D25′—only hb$_3$) and state 10 (hb$_2$ and hb$_3$)—therefore, the additional presence of hb$_3$ compared to the unbound state. However, other hydrogen bond states are also populated, albeit less densely, in the GS. In particular, there is a notable shift towards states containing hb$_1$—for example, states 5–8. There is qualitatively little difference in hydrogen bond distribution between the NAC and GS—the predominant shift with respect to the unbound state is still towards states 2 and 10 as well as a range of other less densely populated states.

There is a lack of hydrogen bond state density for the PR-RT system compared to the other systems. This may be due to the fact that the overall population density of the bound state is very small and therefore due to improper sampling of the hydrogen bond distribution.

Overall, hydrogen bond analysis reveals that NACs do not exhibit a dissimilar hydrogen bond distribution to the GS but the GS in general does promote additional adoption of states involved in the general acid/general base (GA/GB) mechanism for catalysis (specifically hb$_1$ and hb$_3$) as compared to the unbound state. Thus, nucleophilic water binding into the catalytic site—by increasing the probability of forming hb$_3$ in turn promotes formation of hb$_1$, priming the water for successful nucleophilic attack.

## 2.3. Thermodynamic Decomposition of Activation Free Energy Contributions

The free energy of NAC formation was calculated for the enzyme-bound ($\Delta G_N^{\ddagger c}$) and enzyme-free ($\Delta G_N^{\ddagger u}$) substrate systems by computing the mole fraction of NACs compared to the GS in each case. All systems exhibit a relatively small enzyme-bound NAC free energy ($\Delta G_N^{\ddagger c}$) but one which is significant over thermal noise. A range of 2.7 kcal/mol is exhibited, from SP1-NC, the most unfavorable ($\Delta G_N^{\ddagger c}$ = 4.6 ± 0.2 kcal/mol—as reported previously [31]) to TFR-PR, the least unfavorable ($\Delta G_N^{\ddagger c}$ = 1.9 ± 0.2 kcal/mol). By comparison, the enzyme-free NAC free energies ($\Delta G_N^{\ddagger u}$) are of similar overall magnitude but exhibit a smaller range of 1.0 kcal/mol with SP2-p6 the most unfavorable ($\Delta G_N^{\ddagger u}$ = 4.8 ± 0.3 kcal/mol) and MA-CA the least unfavorable ($\Delta G_N^{\ddagger u}$ = 3.8 ± 0.1 kcal/mol).

In order to determine the sensitivity of these results to the distance threshold that partitions the bound from unbound nucleophilic water molecule states, $\Delta G_N^{\ddagger c}$ and $\Delta G_N^{\ddagger u}$ were computed across a range of both $d_n^p$ and $d_o^p$ for each of the enzyme-bound and enzyme-free systems, respectively (Figure 5). All enzyme-bound systems exhibit robust insensitivity to variation in $d_n^p$ between 3.8–5.0 Å. Conversely, all enzyme-free systems expectedly exhibit significant sensitivity to variation in $d_o^p$ in a range of small $d_o^p$ values (3.0–4.5 Å), but then plateau to robust insensitivity for $d_o^p > 4.5$ Å. This further validates the choices of $d_n^p$ and $d_o^p$ for the various systems.

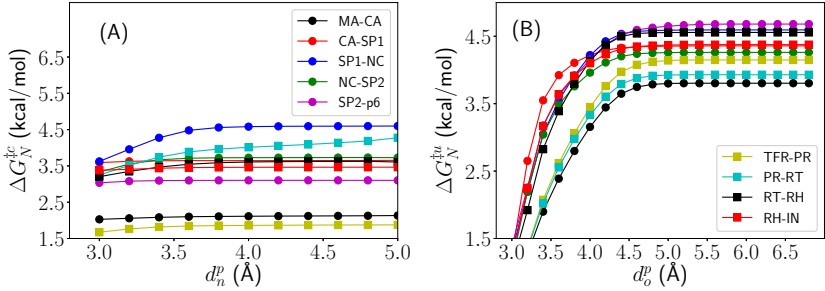

**Figure 5.** Free energy of NAC formation (**A**) for all enzyme-bound ($\Delta G_N^{\ddagger c}$) and (**B**) enzyme-free ($\Delta G_N^{\ddagger u}$) substrate systems when varying the threshold $d_n^p$ and $d_o^p$ that partitions bound and unbound nucleophilic water states, respectively.

The difference in NAC free energy ($\Delta\Delta G_N^{\ddagger}$) between the catalyzed and uncatalyzed systems (Table 2) determines the contribution of NACs to the catalytic effect. These results suggest that NACs contribute only marginally to catalysis in three systems: MA-CA ($-1.7 \pm 0.2$ kcal/mol), SP2-p6 ($-1.7 \pm 0.4$ kcal/mol), and TFR-PR ($-2.2 \pm 0.3$ kcal/mol). In the other systems, the NAC contribution is negligible and/or indistinguishable from thermal noise.

**Table 2.** Activation free energies in both enzyme-catalyzed and uncatalyzed systems decomposed in terms of NAC and non-NAC contributions. All energies are in kcal/mol. Errors for quantities computed in the simulations are shown. Derived and estimated quantities are reported without errors.

| | Uncatalyzed | | | Catalyzed | | | | | |
|---|---|---|---|---|---|---|---|---|---|
| System | $\Delta G^{\ddagger u}$ | $\Delta G_N^{\ddagger u}$ | $\Delta G_{nN}^{\ddagger u}$ | $\Delta G^{\ddagger c\ a}$ | $\Delta G_N^{\ddagger c}$ | $\Delta G_{nN}^{\ddagger c}$ | $\Delta\Delta G^{\ddagger}$ | $\Delta\Delta G_N^{\ddagger}$ | $\Delta\Delta G_{nN}^{\ddagger}$ |
| MA-CA | | $3.8 \pm 0.1$ | 26.2 | 15.7 | $2.1 \pm 0.1$ | 13.6 | −14.3 | $-1.7 \pm 0.2$ | −12.6 |
| CA-SP1 | | $4.4 \pm 0.1$ | 25.6 | 17.0 | $3.7 \pm 0.2$ | 13.3 | −13.0 | $-0.7 \pm 0.3$ | −12.3 |
| SP1-NC | | $4.6 \pm 0.1$ | 25.4 | 15.3 | $4.6 \pm 0.2$ | 10.7 | −14.7 | $0.0 \pm 0.3$ | −14.7 |
| NC-SP2 | | $4.3 \pm 0.1$ | 25.7 | 16.7 | $3.8 \pm 0.2$ | 12.9 | −13.3 | $-0.5 \pm 0.3$ | −12.8 |
| SP2-p6 | $\sim 30\ ^b$ | $4.8 \pm 0.3$ | 25.2 | 18.4 | $3.1 \pm 0.1$ | 15.3 | −11.6 | $-1.7 \pm 0.4$ | −9.9 |
| TFR-PR | | $4.1 \pm 0.1$ | 25.9 | 15.3 | $1.9 \pm 0.2$ | 13.4 | −14.7 | $-2.2 \pm 0.3$ | −12.5 |
| PR-RT | | $3.9 \pm 0.1$ | 26.1 | 15.8 | $4.2 \pm 0.5$ | 11.6 | −14.2 | $0.3 \pm 0.6$ | −14.5 |
| RT-RH | | $4.7 \pm 0.4$ | 25.3 | 17.2 | $3.7 \pm 0.4$ | 13.5 | −12.8 | $-1.0 \pm 0.8$ | −11.8 |
| RH-IN | | $4.4 \pm 0.1$ | 25.6 | 16.6 | $3.5 \pm 0.1$ | 13.1 | −13.4 | $-0.9 \pm 0.2$ | −12.5 |

[a] based on experimentally measured and estimated (see Materials and Methods) $k_{cat}$ values [39–41]; [b] based on an experimentally determined half-life of $\sim$ 500 years [42].

## 3. Discussion

The overall activation energy barrier for uncatalyzed peptide hydrolysis—based on an experimentally determined half-life of $\sim$500 years [42]—is estimated at $\Delta G^{\ddagger u} \sim$30 kcal/mol [31]. Furthermore, as this is likely to be independent of the sequence of flanking amino acids, this value was assigned to all uncatalyzed cleavage substrates in the current study. HIV-1 protease thus exhibits enormous catalytic power, lowering this barrier to $\Delta G^{\ddagger c} \sim$15–18 kcal/mol (a significant reduction of $|\Delta\Delta G^{\ddagger}| \sim$ 12–15 kcal/mol) across the range of substrates it cleaves (Table 2). The results reported here support previous studies [31] and further reinforce the idea that NACs do not contribute significantly to the overall catalytic effect ($|\Delta\Delta G_N^{\ddagger}| \sim$ 0–2 kcal/mol). Rather, non-NAC effects—such as electrostatic preorganization, which is known to play a powerful role in enzyme catalysis [16–22], are thus likely the main source of the catalytic power in HIV-1 protease, contributing a reduction of $|\Delta\Delta G_{nN}^{\ddagger}| \sim$ 10–15 kcal/mol (Table 2). This range is likely to originate from the differing electrostatic properties of the various cleaved peptide sequences and their subsequent differences on the preorganization of the bound enzyme-substrate complex.

The NAC free energy in the uncatalyzed reaction ($\Delta G_N^{\ddagger u}$) varies by only 1.0 kcal/mol in the calculations reported here. Therefore, when taking into account thermal noise ($\sim$0.6 kcal/mol) all systems can be interpreted to have near-identical uncatalyzed NAC free energies, characterized by a mean value of $<\Delta G_N^{\ddagger u}> = 4.3 \pm 0.3$ kcal/mol. Then, assuming the overall uncatalyzed reaction barrier is similar across different sequences, the uncatalyzed non-NAC contribution ($\Delta G_{nN}^{\ddagger u}$) is also nearly identical across the different systems, with a mean value of $<\Delta G_{nN}^{\ddagger u}> = 25.7 \pm 0.3$ kcal/mol.

However, this is not the case in the catalyzed systems. Experimentally, variation of up to three orders of magnitude occurs across $k_{cat}$ [39] for the substrates of HIV-1 protease—from which the above-mentioned variation in catalyzed activation energy barrier is estimated ($\Delta G^{\ddagger c}$(max)–$\Delta G^{\ddagger c}$(min) $\sim$3 kcal/mol). This is similar to the range of variation observed in the computed $\Delta G_N^{\ddagger c}$. If non-NAC contributions were similar to each other in the catalyzed systems, then variation in the overall barrier would also be correlated to variation in $\Delta G_N^{\ddagger c}$. This would be consistent with NACs exhibiting a 'correlative' mode of fine-tuning the catalyzed reaction. However, decomposition into NAC ($\Delta G_N^{\ddagger c}$) and non-NAC ($\Delta G_{nN}^{\ddagger c}$) components shows that both components exhibit a similar range of variation to the overall barrier and $\Delta G^{\ddagger c}$ does not correlate with $\Delta G_N^{\ddagger c}$, exhibiting a correlation coefficient of only $R^2 = 0.01$ (Figure 6A). This implies sequence-dependent variation of both the NAC and non-NAC contributions across the various substrates

in the catalyzed reaction and suggests that differences in activation barrier from cleavage sequence changes are due to the combined and significant variations of both components.

Nonetheless, a subtle role for NACs may still emerge when considering the variation in both components. In the context of the comparably large reductions in energy barrier that non-NAC effects are capable of inducing, their ability to fine-tune such reductions to match the optimally required catalyzed rates across the various substrates may be limited. This is highlighted by comparing non-NAC contributions to the activation barrier with each other (Figure 6B). All the peptide sequences studied here may be grouped into three distinct non-NAC activation energy bands with means at 15.3, 13.3 and 11.2 kcal/mol—where significant gaps (∼2 kcal/mol) exist between the different bands—therefore, no overlap exists when taking into account thermal noise. These bands appear to be too coarsely-tuned to achieve the required differences between the various substrates, whilst the ordering of substrates by non-NAC contributions is not consistent with that of the overall catalyzed energy barriers. Given these non-NAC contributions, invariant NAC contributions in the catalyzed systems, as estimated for the uncatalyzed systems, would resolve neither of these two issues.

Thus, minor variations in the catalyzed NAC free energies ($\Delta G_N^{\ddagger c}$) exhibit a 'complementary' mode of fine-tuning both the differences between and the absolute values of the overall energy barriers to match the optimal requirements for correctly ordered maturation. Furthermore, the existence of a small thermodynamic barrier by NACs enables the catalyzed barrier to reach the virologically relevant region (Figure 6B-yellow band) with the given non-NAC contributions—and thus to meet the overall required timescale for virion maturation.

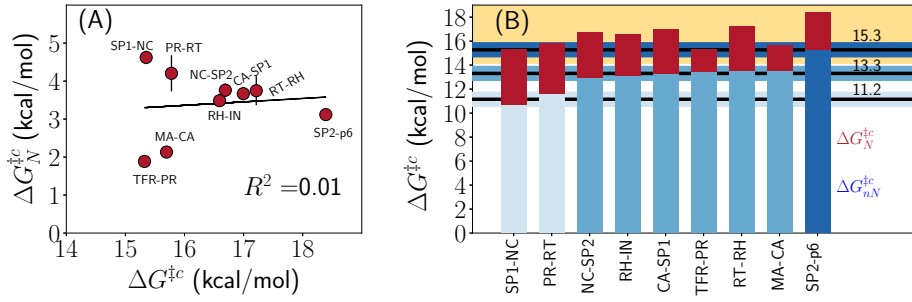

**Figure 6.** (**A**) Plot of experimentally derived values for the activation barrier ($\Delta G^{\ddagger c}$) against calculated values for the free energy of NAC formation in the catalyzed reaction ($\Delta G_N^{\ddagger c}$). (**B**) Comparison of NAC (red), $\Delta G_N^{\ddagger c}$, and non-NAC (blue), $\Delta G_{nN}^{\ddagger c}$, free energy contributions. All systems lie within one of three distinct $\Delta G_{nN}^{\ddagger c}$ bands separated by significantly more than thermal noise (black lines with blue bands). Only one non-NAC band is immediately within the virologically relevant region (yellow) for differential enzyme specificity. NAC contributions are small (< 5 kcal/mol) and within ∼3 kcal/mol of each other, but serve to fine-tune $\Delta G^{\ddagger c}$ for all substrates to both match the optimal barrier differences and reach the relevant barrier region required by the virus—neither of which is achieved, in general, by non-NAC contributions alone.

Why the HIV-1 system has evolved multiple cleavage sequences that result in sometimes very similar $k_{cat}$ is not fully understood. One plausible explanation could be that because sequence change can have an effect on both $k_{cat}$ and $K_M$ and the reaction kinetics of polyprotein cleavage [43,44] is directed by the combination of both, different sequences could control each of these parameters differentially. However, several cleavage sequences recognized by HIV-1 protease both have similar $k_{cat}$ and $K_M$ to each other and are therefore not consistent with the above-mentioned explanation. For example, this applies to the MA-CA (SGNY-PIVQ), SP1-NC (ATIM-MQRG), TFR-PR (SFNF-PQIT), and PR-RT (TLNF-PISP) junctions [39], yet these junctions consist of significantly varying amino acid sequences.

Another possible explanation could be because amino acid sequences can face selection pressure for multiple biomolecular functions not restricted to the given enzymatic reaction. In the case of

polyprotein cleavage by HIV-1 protease, each cleavage site corresponds to the C- and N-termini of its adjunct proteins—several of which are involved in structural and functional interactions beyond cleavage alone. This may impose restrictions on which amino acids are selected in the cleavage sequence—emergent sequences being able to fulfill both the required enzymatic rate as well as the other biomolecular functions. In particular, electrostatic preorganization effects would be highly susceptible to the electrostatic profile of the substrate. Sequence changes selected by other functions may thus correspond to abrupt changes in the enzymatic barrier that are modulated by smaller NAC changes. These large reductions due to non-NAC contributions combined with the fine tuning conferred by NACs may help to fulfill the required enzymatic rate using a different sequence.

Comparing the SP1-NC and TFR-PR substrates serves as a good example to illustrate this. The SP1-NC cleavage site is one of the fastest cleaved sites by HIV-1 protease, sharing the same specificity rate constant as the TFR-PR system [39]. However, it exhibits a non-NAC free energy, which is 2.7 kcal/mol lower than the TFR-PR system (Table 2). The nucleocapsid (NC) protein is essential for condensation of viral RNA [45]—the positively charged arginine in its N-terminus region (MQRG-) is likely to aid non-specific binding to the negatively charged RNA. As part of the SP1-NC cleavage sequence, this residue may also alter the electrostatic balance in the HIV-1 protease active site, contributing to the computed low non-NAC barrier exhibited for this junction. This in turn would lead to mistimed fast cleavage without the additional barrier afforded by the NAC component (4.6 kcal/mol) that restores cleavage rate to the required value. Conversely, the TFR-PR substrate achieves a similar $k_{cat}$ with a different sequence, selected in part due to the structural requirements of the HIV-1 protease N-terminus in forming an interdigitated dimer interface [46] as well as a self-associated precursor protease during intramolecular autocatalysis [47]. This less positive sequence may contribute to the larger non-NAC energy barrier with respect to SP1-NC. However, the same overall energy barrier can still be achieved by this sequence by allowing for a significantly reduced NAC contribution with respect to the enzyme-free system ($\Delta\Delta G_N^{\ddagger} \sim -2$ kcal/mol).

The degree of decoupling between NAC and non-NAC contributions upon single amino-acid mutations has not been investigated here. It is unknown whether a given amino acid mutation in the cleavage region would affect both NAC and non-NAC contributions simultaneously. If so, this would give rise to a more complex picture of fine-tuning where multiple mutations whose sum of NAC and non-NAC contributions kept the desired rate constant. However, single mutations may exist that alter predominantly either NAC or non-NAC contributions. For example, some hydrophobic-hydrophobic mutations may affect the activation barrier more through the NAC rather than the non-NAC mode, whilst hydrophilic-charged changes may alter the non-NAC barrier but not significantly affect catalytic water entry or NAC formation. Both scenarios could easily be accommodated by the fact that $k_{cat}$ depends not only on the immediate residues that juxtapose the lytic bond, but on at least the P4-P4' subsites that constitute the octapeptide cleavage sequence [41].

Future studies that elucidate this might therefore link the role of NAC and non-NAC contributions to the step-wise evolution of cleavage junctions, where sequences that were not initially lytic junctions became so by mutations that first contributed large discrete non-NAC reductions in the energy barrier and whose rate constants were then fine-tuned by subsequent mutations that altered the NAC contribution. This decoupling might also in part account for the role of compensatory mutations [48,49] that restore viral fitness when antiretroviral therapy causes drug resistance mutations deleterious to fitness [50] to emerge.

## 4. Materials and Methods

The role of NAC formation was investigated by performing and comparing ensembles of all-atom explicit solvent molecular dynamics simulations of HIV-1 protease bound to each of a set of eight recognized octapeptide substrates (enzyme-bound) representing inter-protein cleavage junctions (Table 1) in addition to one octapeptide substrate (SP1-NC) that was already previously simulated and

reported [31]. These nine enzyme-bound systems were further compared to corresponding simulations of the apo-ligand (enzyme-free) systems in explicit solvent.

### 4.1. Initial Preparation

Initial structures were taken from the 1KJ4, 1F7A, 1TSU, 1KJF, 1KJ4, 1KJ4, 1KJG, and 1KJH crystal structures of peptide-bound HIV-1 protease complexes for MA-CA, CA-SP1, NC-SP2, SP2-p6, TFR-PR, PR-RT, RT-RH, and RH-IN junctions, respectively [51–53] and, when required, mutated to match the corresponding peptide sequences. Crystallographic water molecules were preserved. An additional water molecule was inserted between the lytic peptide bond and the catalytic dyad as is expected in the general acid/general base (GA/GB) cleavage mechanism [32]. The inactive catalytic dyad D25N was converted into catalytically active D25 form with a monoprotonated state [54,55]. Hydrogen atoms were added, the systems were electrically neutralized (0.15 M NaCl) and explicitly solvated with TIP3P water [56] and topologies were generated using the Leap module of AMBER 14 [57]. The standard AMBER force field for bioorganic systems (ff03) [58] was used to describe all protein parameters. All equilibration and production simulations were performed using ACEMD [59].

### 4.2. Molecular Dynamics Equilibration and Production Simulation Protocol

The molecular dynamics equilibration and simulation protocol for all respective enzyme-bound and enzyme-free systems were identical to those previously reported for the SP1-NC system [31] and are fully described therein. Production ensembles of $100 \times 100$ ns and $10 \times 1$ μs were generated for each enzyme-bound and corresponding enzyme-free apo-ligand system, respectively, in the NVT ensemble with temperature maintained at 300 K. Coordinate snapshots were generated every 100 ps and 10 ps, respectively. Experimental accuracy of the molecular simulation protocol for the HIV-1 protease has been previously validated using NMR $S^2$ order parameters [60]. The effects of forcefield variation are small and have been accounted for in a previous study [31].

### 4.3. Analysis

For the enzyme-bound and enzyme-free systems, the nearest water distance ($d_n$ and $d_o$ respectively) was defined as the distance of the oxygen atom WAT:O of the nearest water molecule from the center of the catalytic site (defined as the geometric center between the four atoms: p1:C, p1′:N, D25:CG and D25′:CG) and the center of the lytic peptide bond (p1:C-p1′:N), respectively (Figure 1). The ground state (GS) was characterized by the nearest water molecule being within an appropriate distance cutoff ($d_n^p$ and $d_o^p$) from the respective centers in the enzyme-bound and enzyme-free systems. These thresholds were chosen after analysis of the density distributions for nucleophilic water binding (see Results). The nucelophile attack distance ($d_a$) was defined as the distance between the carbon atom of the lytic peptide bond and the oxygen atom of the nearest water molecule (WAT$_{nuc}$:O$_{nuc}$-p1:C). The nucelophile attack angle ($\alpha$) was defined as the angle between the $d_a$ vector and the vector corresponding to the carbonyl bond adjacent to the lytic peptide bond (WAT$_{nuc}$:O$_{nuc}$-p1:C-p1:O). NACs were characterized in terms of Bürgi–Dunitz criteria corresponding to an angle range of $100° \leq \alpha \leq 110°$ and distance threshold of $d_a \leq 3.2$ Å. A hydrogen bond network was analyzed in the catalytic site of the enzyme-bound systems—characterized by cooperative combinations of a set of four distinct hydrogen bonds $hb = \{ hb_1, hb_2, hb_3, hb_4 \}$ and resulting in a set of 12 non-zero density hydrogen bond states $s = \{1,...,12\}$ (Figure 4). The threshold for a hydrogen bond was a donor–acceptor distance $\leq 3.5$ Å and donor–hydrogen-acceptor angle of $\geq 150°$.

Probability densities ($\rho(d_n)$ and $\rho(d_o)$) were calculated by binning ensemble data along the corresponding reaction coordinate space, respectively ($d_n$ and $d_o$) using kernel density estimation with an Epanechnikov kernel and bandwidth parameter $h = 0.75$. The mole fraction ($\Gamma_M$) of a given macrostate ($M$) was computed as $\Gamma_M = m/N$, where $m$ is the number of datapoints within the above-mentioned boundary partitions of the macrostate in the corresponding reaction coordinate space, and $N$, the total number of datapoints in the given ensemble. The potential of mean force

for a given state ($G_M$) was calculated as $G_M = -k_B T ln(\Gamma_M)$. Free energy differences ($\Delta G$) between various macrostates were calculated from the ratios of the corresponding mole fractions according to $\Delta G = -k_B T ln(\Gamma_{M2}/\Gamma_{M1})$, where $\Gamma_{M1}$ and $\Gamma_{M2}$ are mole fractions of any given states $M1$ and $M2$, respectively. Hydrogen bond state mole fractions ($\Gamma_s$) within a given macrostate were calculated as $\Gamma_s = n_s/m$, where $n_s$ is the number of datapoints within the criteria for each hydrogen bond state $s$ within the subset of datapoints ($m$) comprising state $M$. Care was taken to integrate the possible degenerate configurations pertaining to each distinct hydrogen bond that arise from structural symmetries imposed by molecular rotation—as previously reported [31].

The complete ensemble for each system was used to perform the hydrogen bond (Figure 4) and free energy variation (Figure 5) analyses. Reported free energy values, as well as their errors (Tables 1 and 2), were calculated by partitioning each ensemble into five subsets in both the enzyme-bound and enzyme-free systems; mole fractions were calculated independently in each subset and averaged to yield means and standard deviations. The only exception to this was the calculation of the NAC free energy for the enzyme-bound PR-RT system for which very few absolute counts were exhibited—thus the ensemble was divided into only two subsets to compute means and errors here.

Estimates for the overall catalyzed energy barrier ($\Delta G^{\ddagger c}$) for all substrates except NC-SP2, were made based on converting experimental $k_{cat}$ values [39] using Equation (2) and assuming $\gamma = 1$ at 300 K. NC-SP2 was not measured in [39], but was measured in a similar study at 310 K [41], as were several other cleavage junctions at this temperature [40]. Therefore, $k_{cat}$ for NC-SP2 at 300 K was estimated by multiplying the $k_{cat}$ ratio of NC-SP2/CA-SP1 from [40,41] to the $k_{cat}$ value of CA-SP1 in the main set [39] from which $\Delta G^{\ddagger c}$ was subsequently calculated.

## 5. Conclusions

Computational studies have previously revealed the existence of a small but significant thermodynamic barrier contributed by the formation of near attack conformations (NACs) that lie on the transition path of the peptide hydrolysis reaction catalyzed by HIV-1 protease [31]. However, NACs were also found there to confer no catalytic effect because the thermodynamic barrier to their formation was equivalent in the uncatalyzed reaction. In the current study, the role of NACs has been further explored across a range of substrates cleaved by the HIV-1 protease, using the same all-atom ensemble molecular dynamics simulation approach coupled to Bürgi–Dunitz theory for characterizing nucleophilic attack of a water molecule on the lytic peptide bond.

This study supports the previous findings that NACs play little or no role in the catalytic effect induced by HIV-1 protease—catalytic barrier reduction is thus entirely dominated by non-NAC contributions. Nonetheless, the functional role of NACs may emerge when considering them together with non-NAC contributions as well as the virological requirements for the ordering of cleavage rates across the different substrates. For HIV-1, the kinetic order of cleavage is tightly regulated [61] to achieve correct architectural reorganization into a mature virion [62] within the physiologically required timescale for infection [43,44].

The findings reported here suggest that NAC contributions, whilst small, are also largely invariant across multiple substrates in the uncatalyzed reaction. Similarly, non-NAC contributions, even though large, are also relatively invariant because the uncatalyzed barrier may be independent of peptide sequence. The catalyzed reactions, however, present a different picture because a range of variation (~3 kcal/mol) exists in the overall energy barrier and both non-NAC and NAC contributions are shown to vary by ~3 kcal/mol across the range of substrates studied. Although non-NAC contributions dominate the reduction in the overall barrier, they appear to be grouped in discrete energy bands that are too coarsely separated to account for either the small variations or the correct ordering of the overall barriers across the substrates. The small variation in NAC contributions thus may constitute a complementary fine-tuning mechanism to rectify both of these issues.

Biological systems face evolutionary selection pressure on multiple fronts and viruses, in particular, are well-known for their parsimony in achieving macromolecular functionality. Together,

such a combination of coarse-reduction by non-NAC and complementary fine-tuning by NAC contributions, respectively, may thus provide a way for controlling required enzymatic rates, whilst under sequence selection pressure for other biomolecular functions. There is likely a trade-off in selecting amino-acid sequences for these functions and those required to directly optimize the given enzymatic reaction. This selection pressure may make it difficult to fine-tune the necessary differences in enzymatic rate by solely using changes in non-NAC contributions. NACs offer an alternative mode to accommodate such differences whilst modulating the rate to meet that required biologically for optimal processing at the corresponding site.

Establishing NAC effects have proven computationally challenging because rigorous characterization at the limit of a classical forcefield requires sufficient sampling to observe ground state (GS) motions towards the transition state. This has challenged the exploration of relative NAC effects in similar but non-identical systems. Here, sufficient sampling of the GS is made possible for most systems by reversible water entry to the active-site with around 10 µs of aggregate sampling per enzyme-bound system. Other systems of interest for exploring NAC contributions may therefore be accessible with current computational power.

Furthermore, future studies using quantum mechanical/molecular mechanics (QM/MM) approaches [22] may test the accuracy of the classical observations exhibited here in more detail and also establish a quantum analog of the nucleophilic attack criteria given by Bürgi–Dunitz theory, from which NAC contributions could be better decoupled from subsequent steps along the reaction pathway.

Nonetheless, the observation here that, for the HIV-1 protease system, relative enzyme specificity is in part directed by fine-tuned nucleophilic water NAC contributions to the catalyzed reaction barrier, implies that similar NAC-specificity effects may exist in other enzyme–substrate reactions and suggests a novel mechanism for fine-tuning and controlling such reactions.

**Funding:** The author acknowledges support from amfAR Mathilde Krim Fellowship in Basic Biomedical Research number 108680, from the Volkswagen Foundation 'Experiment! Funding Initiative' Grant No. 93874 and from the Klaus Tschira Foundation.

**Acknowledgments:** The author thanks Gianni De Fabritiis for previous use of the GPUGRID.net infrastructure as well as Peter Coveney, Natalia Gabrielli, Andreas Meyerhans, Gilles Mirambeau and Sébastien Lyonnais for valuable discussions.

**Conflicts of Interest:** The author declares no conflict of interest.

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
