# Peer review of "Fine-Tuning of Sequence Specificity by Near Attack Conformations in Enzyme-Catalyzed Peptide Hydrolysis"

_catalysts, doi:10.3390/catal10060684_

Round 1

Reviewer 1 Report

The work presented here is delightful. It is clearly demonstrated that non-near attack conformation effects contribute largely to the reduction of activation energy barrier in HIV-1 protease with a range of substrates. While near attack conformation contributions are small, they serve to fine tune the activation energy barrier. Some minor suggestions and one question:

  1. The format of kcat should be revised with ‘k’ being italic while subscript ‘cat’ not being italic.

  1. In table 1, free energies of nucleophilic water binding were presented. Is there any correlation between nucleophilic water binding energies with kcat (ΔG‡c) with a range of substrates, or nucleophilic water binding is preorganized in the active site thus not reflected in kcat (ΔG‡c ) ?

  1. In table 2, ‘Catalyzed’ in line 1 is misplaced.

  1. Line 220, there is a redundant ‘the’.

Author Response

I am grateful to the reviewer for reviewing the manuscript and for the helpful suggestions. Please find my response to each suggestion below:

­­­ The work presented here is delightful. It is clearly demonstrated that non-near attack conformation effects contribute largely to the reduction of activation energy barrier in HIV-1 protease with a range of substrates. While near attack conformation contributions are small, they serve to fine tune the activation energy barrier. Some minor suggestions and one question:

1. The format of kcat should be revised with ‘k’ being italic while subscript ‘cat’ not being italic.

This has now been corrected throughout the manuscript.

2. In table 1, free energies of nucleophilic water binding were presented. Is there any correlation between nucleophilic water binding energies with kcat (ΔG‡c) with a range of substrates, or nucleophilic water binding is preorganized in the active site thus not reflected in kcat (ΔG‡c ) ?

There is a very weak negative correlation between the nucleophilic water binding free energy and the catalyzed free energy barrier (R2=0.43) but as the reviewer pointed to, indeed the nucleophilic water binding is a minor second step to achieve the ground state (GS) from the fully unbound state:

E + S + H2O -> E.S + H2O ->  E.GS

and thus considered here decoupled from the subsequent E.GS->E.TS step to the transition state (TS), represented by kcat .

3. In table 2, ‘Catalyzed’ in line 1 is misplaced.

Thanks for spotting this formatting error. The line intended to say Non-Catalyzed followed by Catalyzed and has now been corrected.

4. Line 220, there is a redundant ‘the’.

This has now been corrected.

Reviewer 2 Report

The manuscript “Fine-Tuning of Sequence Specificity by Near Attack Conformations in Enzyme-Catalyzed Peptide Hydrolysis” by S. Kashif Sadiq describes the computational analysis regarding the role of near attack conformations on the catalytic rates of an HIV protease to cleave nine different peptides. It represents a seamless extension of a previous study of the same author and covers an interesting question that deserves attention. There are a few issues that need to be addressed:

The manuscript is well written, however several of the figures shown in the manuscript are taken from the author’s previous paper without adequate referencing. This includes figures 1 and figures 4A and 4B.   

The author shows in figure 6A that there is no correlation between the NAC contributions and the specificity of HIV protease toward different substrates. The author should clarify how these contributions might fine-tune the peptide cleavage reaction if there is no correlation with the specificities.

Likewise, the study does not provide an explanation of how the NACs might fine-tune the different catalytic rates of the various peptide substrates. The cleavage sites of TFR-PR and PR-RT are both flanked by the same residues (phenylalanine and proline) and exhibit very similar ΔG‡c based on experimental results. According to the present study, they have very different NAC and non-NAC contributions, which imply that the NACs play a minor role for the catalytic reaction.

In Material and Methods, more information should be given what parameters were used for microstate binning to compute PMFs.

Author Response

I am grateful to the reviewer for reviewing the manuscript and for the helpful comments. Please find my response to each comment below:

The manuscript “Fine-Tuning of Sequence Specificity by Near Attack Conformations in Enzyme-Catalyzed Peptide Hydrolysis” by S. Kashif Sadiq describes the computational analysis regarding the role of near attack conformations on the catalytic rates of an HIV protease to cleave nine different peptides. It represents a seamless extension of a previous study of the same author and covers an interesting question that deserves attention. There are a few issues that need to be addressed:

The manuscript is well written, however several of the figures shown in the manuscript are taken from the author’s previous paper without adequate referencing. This includes figures 1 and figures 4A and 4B.    

Figures 1, 4A and 4B are all explanatory figures, rather than those that present novel data. Figure 1 is a more detailed adaptation of a similar figure in the previous paper whilst Figures 4A and 4B, as the reviewer correctly points out, have been used here again. I have now referenced the previous paper in the captions for the stated figures.

The author shows in figure 6A that there is no correlation between the NAC contributions and the specificity of HIV protease toward different substrates. The author should clarify how these contributions might fine-tune the peptide cleavage reaction if there is no correlation with the specificities.

The Discussion section has now been expanded to clarify different types of fine-tuning. The reviewer is indeed correct that ‘correlative’ fine-tuning is one potential mode – which would be the case if the catalyzed non-NAC contributions would all be invariant and if NAC contributions would correlate with the overall barrier. These systems seem to exhibit a different kind of fine-tuning which we have termed ‘complementary’ fine-tuning due to the fact that the non-NAC contributions are not invariant and moreover are coarsely separated – therefore alone cannot account for the small differences in overall cleavage required to regulate the reaction kinetics of polyprotein cleavage. NACs provide a means to fine-tune these relative differences because they are not as strong as non-NAC contributions.

Likewise, the study does not provide an explanation of how the NACs might fine-tune the different catalytic rates of the various peptide substrates. The cleavage sites of TFR-PR and PR-RT are both flanked by the same residues (phenylalanine and proline) and exhibit very similar ΔG‡c based on experimental results. According to the present study, they have very different NAC and non-NAC contributions, which imply that the NACs play a minor role for the catalytic reaction.

The current study indeed confirms that NACs do not play a significant role in the overall catalytic reaction. This has been stated in the text.  A Conclusion section has now also been added in which this is further stated.   The catalytic power afforded by NACs across all the substrates considered is not more than ~2 kcal/mol. However, the additional and separate finding is that NACs do play a role in making small alterations in order to match the required rates for polyprotein cleavage.  In order for the reaction kinetics to occur at all the HIV-1 protease has to reduce the barrier from ~30 kcal/mol to in the region of 15-18 kcal/mol.  NACs do not contribute significantly to this massive barrier reduction. But in addition to this, all the cleavage sites need to have the relative barriers they do because if they didn’t virion maturation would not occur correctly and would result in aberrant virion morphologies. These barriers are very close to each other, sometimes within 1 kcal/mol – and that is where it is suggested small changes in NACs can help fine-tune the larger differences between the non-NACs.

The Discussion section has also now been expanded to suggest possible explanations for both the sequence variation and possible fine-tuning by NACs. It is indeed an interesting question why different recognized cleavage sequences have evolved that exhibit the same ΔG‡c  . Two possible explanations are considered. One is that KM is also important in regulating the overall reaction kinetics as well as kcat.  Therefore, different sequences could have an effect on kcat and KM differentially, for example, keeping kcat the same and altering KM. However, four cleavage sequences (including MA-CA, SP1-NC, PR-RT and TFR-PR) have both very similar kcat and KM experimentally but still have varying sequences. So, the previously offered explanation of sequence selection due to a combined requirement for a given enzymatic rate as well as fulfilling other molecular functions is likely for these cases and is retained in the Discussion.

The TFR-PR and PR-RT do indeed have very different NAC and non-NAC contributions.  PR-RT is very similar in NAC and non-NAC contributions to SP1-NC.  The original manuscript compared TFR-PR with SP1-NC precisely for this reason that the same overall energy barrier can come from different combinations of NAC and non-NAC.  Both SP1-NC and PR-RT are substantially different to TFR-PR. This is because even though TFR-PR and PR-RT have the same flanking residues at the P1 and P1’ positions, sequence recognition occurs at least across the whole octapeptide e.g. P4-P4’ subsites. Therefore, the further flanking residues are also important and are not the same across these substrates.  A paragraph has been added to elaborate how single amino acid changes may have more of an effect on either the NAC or non-NAC contributions. This is beyond the remit of the current study, but future studies could look at stepwise mutations and compute whether these changed NAC, non-NAC or both.

In Material and Methods, more information should be given what parameters were used for microstate binning to compute PMFs.

The Analysis section of the Material and Methods has been expanded to more formally describe the density estimation procedure involved in computing the probability density functions along the reaction coordinate space as well as the partitioning of the data within different boundary conditions related to the difference macrostates, h-bond states, mole fraction computation, free energies and PMFs.

Round 2

Reviewer 2 Report

This is a revised manuscript and the author has responded carefully to the raised issues. I have no further critical comments.